# DISTRIBUTIONAL ADVERSARIAL NETWORKS

## ABSTRACT

In most current formulations of adversarial training, the discriminators can be expressed as single-input operators, that is, the mapping they define is separable over observations. In this work, we argue that this property might help explain the infamous mode collapse phenomenon in adversarially-trained generative models. Inspired by discrepancy measures and two-sample tests between probability distributions, we propose *distributional* adversaries that operate on samples, i.e., on sets of multiple points drawn from a distribution, rather than on single observations. We show how they can be easily implemented on top of existing models. Various experimental results show that generators trained in combination with our distributional adversaries are much more stable and are remarkably less prone to mode collapse than traditional models trained with observation-wise prediction discriminators. In addition, the application of our framework to domain adaptation results in strong improvement over recent state-of-the-art.

## 1 INTRODUCTION

Adversarial training of neural networks, especially Generative Adversarial Networks (GANs) (Goodfellow et al., 2014), has proven to be a powerful tool for learning rich models, leading to outstanding results in various tasks such as realistic image generation, text to image synthesis, 3D object generation, and video prediction (Reed et al., 2016; Wu et al., 2016; Vondrick et al., 2016). Despite their success, GANs are known to be difficult to train. The generator and discriminator can oscillate significantly from iteration to iteration, and slight imbalances in their capacities frequently cause the training objective to diverge. Another common problem suffered by GANs is *mode collapse*, where the distribution learned by the generator concentrates on a few modes of the true data distribution, ignoring the rest of the space. In the case of images, this failure results in generated images that albeit realistic, lack diversity and reduce to a handful of prototypes.

A flurry of recent research seeks to understand and address the causes of instability and mode collapse in adversarially-trained models. The first insights come from Goodfellow et al. (2014), who note that one of the main causes of training instability is saturation of the discriminator. Arjovsky & Bottou (2017) formalize this idea by showing that if the two distributions have supports that are disjoint or concentrated on low-dimensional manifolds that do not perfectly align, then there exists an optimal discriminator with perfect classification accuracy almost everywhere and the usual divergences (Kullback-Leibler, Jensen-Shannon) max-out for this discriminator. In follow-up work, Arjovsky et al. (2017) propose an alternative training scheme (WGAN) based on estimating the Wasserstein distance instead of the Jensen-Shannon divergence between real and generated distributions.

In this work, we highlight a further view on mode collapse. The discriminator part of GANs and of variations like WGANs is separable over observations, which, as we will illustrate, can result in serious problems, even when minibatches are used. The underlying issue is that the stochastic gradients are essentially (sums of) functions of *single* observations (training points). Despite connections to two-sample tests based on Jensen-Shannon divergence, ultimately the updates based on gradients from different single observations are completely independent of each other. We show how this lack of sharing information between observations may explain mode collapses in GANs.

Motivated by these insights, we take a different perspective on adversarial training and propose a framework that brings the discriminator closer to a truly *distributional* adversary, i.e., one that

considers a sample[1] (a *set* of observations) in its entirety, retaining and sharing global information between gradients. The key insight is that a carefully placed nonlinearity in the form of specific population comparisons can enable information-sharing and thereby stabilize training. We develop and test two such models, and also connect them to other popular ideas in deep learning and statistics.

**Contributions.** The main contributions of this work are as follows:

• We introduce a new *distributional* framework for adversarial training of neural networks that operates on a genuine *sample*, i.e., a collection of points, rather than an observation. This choice is *orthogonal* to modifications of the types of loss (e.g. logistic vs. Wasserstein) in the literature.

• We show how off-the-shelf discriminator networks can be made *distribution-aware* via simple modifications to their architecture and how existing models can seamlessly fit into this framework.

• Empirically, our distributional adversarial framework leads to more stable training and significantly better mode coverage than common single-observation methods. A direct application of our framework to domain adaptation results in strong improvements over state-of-the-art.

## 2 FAILURE OF SINGLE-OBSERVATION DISCRIMINATORS

To motivate our distributional approaches, we illustrate with the example of the original GAN how training with single-observation-based adversaries might lead to an *unrecoverable* mode collapse in the generator. The objective function for a GAN with generator $G$ and discriminator $D$ is

$$\min_G \max_D V(G, D) = \min_G \max_D \left\{ \mathbb{E}_{x \sim \mathbb{P}_x}[\log D(x)] + \mathbb{E}_{z \sim \mathbb{P}_z}[\log(1 - D(G(z)))] \right\}, \qquad (2.1)$$

where $D : \mathbb{R}^d \to [0, 1]$ maps an observation to the probability that it comes from data distribution $\mathbb{P}_x$, and $G : \mathbb{R}^l \to \mathbb{R}^d$ is a network parametrized by $\theta_G$ that maps a noise vector $z \in \mathbb{R}^l$, drawn from a simple distribution $\mathbb{P}_z$, to the original data space. The aim of $G$ is to make the distribution $\mathbb{P}_G$ of its generated outputs indistinguishable from the training distribution.

In practice, $G$ is trained to instead maximize $\log D(G(z))$ to prevent loss saturation early in the training. Goodfellow et al. (2014) showed that the discriminator converges in the limit to $D^*(x) = \mathbb{P}_x(x)/(\mathbb{P}_x(x) + \mathbb{P}_G(x))$ for a fixed $G$. In this limit, given a sample $Z = \{z^{(1)}, \ldots, z^{(B)}\}$ from the noise distribution $\mathbb{P}_z$, the gradient of $G$'s loss with respect to its parameters $\theta_G$ is

$$\nabla_{\theta_G} \text{loss}(Z) = \frac{1}{B} \sum_{i=1}^{B} \frac{1}{D(G(z^{(i)}))} \nabla D(G(z^{(i)})) \left[ \frac{d}{d\theta_G} G(z^{(i)}) \right], \qquad (2.2)$$

where we slightly abuse the notation $\frac{d}{d\theta_G} G$ to denote $G$'s Jacobian matrix. Thus, the gradient with respect to each observation is weighted by terms of the form $\nabla D(x_G^{(i)})/D(x_G^{(i)})$, where $x_G^{(i)} := G(z^{(i)})$. These terms can be interpreted as relative slopes of the discriminator's confidence function. Their magnitude and sign depend on $\frac{dD}{dx_G}$ (the slope of $D$ around $x_G$) and on $D(x_G)$, the confidence that $x_G$ is drawn from $\mathbb{P}_x$. In at least one notable case this ratio is unambiguously low: values of $x$ where the discriminator has high confidence of real samples *and* flat slope. This implies that given the definition of $D^*$, the weighting term will vanish in regions of the space where the generator's distribution has constant, low probability compared to the real distribution, such as neighborhoods of the support of $\mathbb{P}_x$ where $\mathbb{P}_G$ is missing a mode. Figure 1 exemplifies this situation for a simple case in 1D where the real distribution is bimodal and the generator's distribution is currently concentrated around one of the modes. The cyan dashed line is the weighting term $\nabla D(x_G)/D(x_G)$, confirming our analysis above that gradients for points around the second mode will vanish.

The effect of this vanishing of mode-seeking gradients during training is that it prevents $G$ from spreading mass to other modes, particularly distant ones. Whenever $G$ does generate a point in a region far away from where most of its mass is concentrated (an event that by definition already occurs with low probability), this example's gradient (which would update $G$'s parameters to move mass to this region) will be heavily down-weighted and therefore dominated by high-weighted gradients of other examples in the batch, such as those in spiked high-density regions. The result is a catastrophic *averaging-out*. A typical run of a GAN suffering from this phenomenon on a simple distribution is shown in Figure 2 (top row), where the generator keeps generating points from the same mode across the training procedure and is unable to recover from mode collapse.

---

[1] Throughout this paper we will use the term *sample* to refer to a set of instances generated from a distribution, and *observation* to refer to an element of that sample.

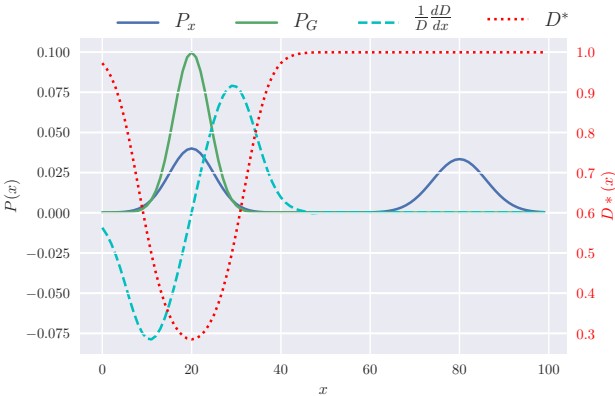

Figure 1: Intuition behind mode-collapsing behavior in single-observation discriminators with logistic loss. The current generated distribution (solid green line) covers only one of the true distribution's modes. Gradients with respect to generated points $x$ are weighted by the term $-\frac{1}{D}\frac{dD}{dx}$ (cyan dashed line), so gradients corresponding to points close to the second mode will be dominated by those coming from the first mode.

At a high level, this mode-collapsing phenomenon is a consequence of a myopic discriminator that bases its predictions on a single observation, leading to gradients that are not harmonized by global information.[2] Discriminators that instead predict based on an entire sample are a natural way to address this issue (Figure 2, bottom two rows), as we will show in the remainder of this work.

## 3  DISTRIBUTIONAL APPROACHES TO ADVERSARIAL TRAINING

To mitigate the above noted failure, we seek a discriminator that considers a sample (multiple observations) instead of a single observation when predicting whether the input is drawn from $\mathbb{P}_x$. More concretely, our discriminator is a *set function* $M : 2^{\mathbb{R}^d} \to \mathbb{R}$ that operates on a sample $\{x^{(1)}, \dots, x^{(B)}\}$ of potentially varying size. Next, we construct this discriminator step by step.

### 3.1  DISTRIBUTIONAL ADVERSARIES

**Neural Mean Embedding.**  Despite its simplicity, the mean is a surprisingly useful statistic for discerning between distributions, and is central to the *Maximum Mean Discrepancy* (MMD) (Gretton et al., 2005; Fukumizu et al., 2008; Smola et al., 2007), a powerful discrepancy measure between distributions that enjoys strong theoretical properties. Instead of designing a kernel explicitly as in MMD, here we learn it in a fully data-driven fashion. Specifically, we define a *neural mean embedding* (NME) $\eta$, where $\phi$ is learned as a neural network:

$$\eta(\mathbb{P}) = \mathbb{E}_{x \sim \mathbb{P}}[\phi(x)]. \tag{3.1}$$

In practice, $\eta$ only has access to $\mathbb{P}$ through samples of finite size $\{x^{(i)}\}_{i=1}^B \sim \mathbb{P}$, thus one effectively uses the empirical estimate

$$\widehat{\eta}(X = \{x^{(1)}, \dots, x^{(B)}\}) = \tfrac{1}{B} \sum\nolimits_{i=1}^B \phi(x^{(i)}). \tag{3.2}$$

This distributional encoder forms one pillar of our adversarial learning framework. We propose two alternative adversary models that build on this NME to discriminate between samples and produce a rich training signal for the generator.

**Sample Classifier.**  First, we combine the NME with a classifier to build a discriminator for adversarial training. That is, given an empirical estimate $\widehat{\eta}(X)$ from a sample $X$ (drawn from the real data $\mathbb{P}_x$ or the generated distribution $\mathbb{P}_G$), the classifier $\psi_S$ outputs 1 to indicate the sample was drawn from $\mathbb{P}_x$ and 0 otherwise. Scoring the discriminator's predictions via the logistic loss leads to the following value function for the adversarial game:

$$V_S(G, D_S) = \log(D_S(X)) + \log(1 - D_S(G(Z))), \tag{3.3}$$

where $X$ and $Z$ are samples from $\mathbb{P}_x$ and $\mathbb{P}_z$ respectively, and $D_S(\cdot) = \psi_S \circ \widehat{\eta}(\cdot)$ is the full discriminator (NME and classifier), which we refer to as the *sample classifier*. Eq. (3.3) is similar

---

[2]Note that even if training with minibatches, the aggregation happens *after* $D$'s computation, preventing harmonization. This is not the case for the adversaries proposed in Section 3.

to the original GAN objective (2.1), but differs in a crucial aspect: here, the expectation is *inside* $\psi_S$, and the classifier is only predicting *one* label for the entire sample instead of one label for each observation. In other words, while in a GAN $D$ operates on single observations and then aggregates its predictions, here $D_S$ first aggregates the sample and then operates on this aggregated representation.

**Two-sample Discriminator.** Inspired by two-sample tests, we alternatively propose to shift from a classification to a *discrepancy* objective, that is, given two samples $X, Z$ drawn independently, the discriminator predicts whether they are drawn from the same or different distributions. Concretely, given two NMEs $\eta(\mathbb{P}_1)$ and $\eta(\mathbb{P}_2)$, the two-sample discriminator $\psi_{2S}$ uses their absolute difference to output $\psi_{2S}(|\eta(\mathbb{P}_1) - \eta(\mathbb{P}_2)|) \in [0, 1]$, interpreted as the confidence that the two samples were indeed drawn from different distributions. Again with the logistic loss, we arrive at the objective function

$$V_{2S}(G, D_{2S}) = \log(D_{2S}(G(Z), X)) + \\ \frac{1}{2}\left(\log(1 - D_{2S}(G(Z_1), G(Z_2))) + \log(1 - D_{2S}(X_1, X_2))\right), \tag{3.4}$$

where we split each of the two samples $X$ and $Z$ into halves ($X_1, X_2$ from $X$ and $Z_1, Z_2$ from $Z$), and the second line evaluates the discrepancy between the two parts of the same sample, respectively. As before, we use $D_{2S}(\cdot, \cdot) = \psi_{2S} \circ |\widehat{\eta}(\cdot) - \widehat{\eta}(\cdot)|$ to denote the full model and refer to it as a *two-sample discriminator*.

**Choice of Objective Function.** Despite many existing adversarial objective functions in the literature (Mao et al., 2016; Arjovsky et al., 2017; Mroueh & Sercu, 2017; Bellemare et al., 2017; Li et al., 2017; Mroueh et al., 2017), here we use the logistic loss to keep the presentation simple, to control for varying factors and to extract the effect of just our main contribution: namely, that the adversary is *distribution-based* and returns only one label per sample, a crucial departure from existing, single-observation adversary models such as the well-understood vanilla GAN. This choice is *orthogonal* to modifications of the types of loss (e.g., logistic vs. Wasserstein). It can, indeed, be seamlessly combined with most existing models to enhance their ability to match distributions.

### 3.2 DISTRIBUTIONAL ADVERSARIAL NETWORKS

We use the novel distributional adversaries proposed above in a new training framework that we name *Distributional Adversarial Network (DAN)*. This framework can easily be combined with existing adversarial training algorithms by a simple modification of their adversaries. In this section, we examine in detail an example application of DAN to the generative adversarial setting. In the experiments section we provide an additional application to adversarial domain adaptation.

Using distributional adversaries within the context of GANs yields a saddle-point problem analogous to the original GAN (2.1):

$$\min_G \max_{D_\xi} V_\xi(G, D_\xi), \tag{3.5}$$

where $\xi \in \{S, 2S\}$ and the objective function $V_\xi$ is either (3.3) or (3.4); we refer to these as DAN-$\xi$. As with GAN, we can optimize (3.5) via alternating updates on $G$ and $D_\xi$.

Although optimizing (3.5) directly yields generators with remarkable performance in simple settings (see empirical results in Section 5.1), when the data distribution is complex, the distributional adversaries proposed here can easily overpower $G$, particularly early in the training when $\mathbb{P}_G$ and $\mathbb{P}_x$ differ substantially. Thus, we propose to optimize instead a *regularized* version of (3.5):

$$\min_G \max_{D, D_\xi} V_\xi(G, D_\xi) + \lambda V(G, D), \tag{3.6}$$

where $D$ is a weaker single-observation discriminator (such as the one in the original GAN), and $\lambda$ is a parameter that trades off the strength of this regularization. Note that for $\lambda \to \infty$ we recover the original GAN objective, while $\lambda = 0$ yields the *purely distributional* loss (3.5). In between these two extremes, the generator receives *both local and global* training signals for every generated sample. We show empirically that DAN training is stable across a reasonable range of $\lambda$ (Appendix B).

During training, all expectations with respect to data and noise distributions are approximated via finite sample averages. In each training iteration, we draw samples from the data and noise

distributions. While for DAN-S the training procedure is similar to that of GAN, DAN-2S requires a modified training scheme. Due to the form of the two-sample discriminator, we want a balanced exposure to pairs of samples drawn from the same and different distributions. Thus, every time we update $D_{2S}$, we draw samples $X = \{x^{(1)}, \ldots, x^{(B)}\} \sim \mathbb{P}_x$ and $Z = \{z^{(1)}, \ldots, z^{(B)}\} \sim \mathbb{P}_z$ from data and noise distributions. We then split each sample into two parts, $X_1 := \{x^{(i)}\}_{i=1}^{\frac{B}{2}}$, $X_2 = \{x^{(i)}\}_{i=\frac{B}{2}+1}^{B}$, $Z_1 := \{z^{(i)}\}_{i=1}^{\frac{B}{2}}$, $Z_2 = \{z^{(i)}\}_{i=\frac{B}{2}+1}^{B}$ and use the discriminator $D_{2S}$ to predict on each pair of $(X_1, G(Z_2))$, $(G(Z_1), X_2)$, $(X_1, X_2)$ and $(G(Z_1), G(Z_2))$ with target outputs 1, 1, 0 and 0, respectively. A detailed training procedure and its visualization are shown in Appendix A.

### 3.3 Gradient weight sharing in distributional adversaries

We close this section discussing why the distributional adversaries proposed above overcome the vanishing gradients phenomenon described in Section 2. In the case of the sample classifier ($\xi = S$), a derivation as in Section 2 shows that for a sample $Z$ of $B$ points from the noise distribution, the gradient of the loss with respect to $G$'s parameters is

$$\nabla_{\theta_G} \text{loss}(Z) = \frac{1}{\psi_S(\widehat{\eta}_B)} \nabla \psi_S(\widehat{\eta}_B) \left( \frac{1}{B} \sum_{i=1}^{B} \nabla \psi_S(G(z^{(i)})) \left[ \frac{d}{d\theta_G} G(z^{(i)}) \right] \right) \qquad (3.7)$$

where we use $\widehat{\eta}_B := \widehat{\eta}(\{G(z^{(1)}), \ldots, G(x^{(B)})\})$ for ease of notation. Note that, as opposed to (2.2), the gradient for each variable $z^{(i)}$ is weighted by the same left-most discriminator confidence term. This has the effect of *sharing information across observations* when computing gradients: whether a sample (encoded as a vector $\widehat{\eta}_B$) can fool the discriminator or not will have an effect on every observation's gradient. The benefit is clearly revealed in Figure 2 (bottom two rows), where, in contrast to the vanilla GAN which remains stuck in mode collapse, DAN is able to recover all modes.

The gradient (3.7) suggests that the true power of this sample-based setting lies in choosing a discriminator $\psi_S$ that, through non-linearities, enforces interaction between the points in the sample. The notion of sharing information across examples occurs also in batch normalization (BN) (Ioffe & Szegedy, 2015), although the mechanism to achieve this interaction and the underlying motivation for doing it are very different. While the analysis here is not rigorous, the intuitive justification for sample-based aggregation is clear, and is confirmed by our experimental results.

## 4 Connections

**Distributional Adversaries as Discrepancy Measures.** Our sample classifier implicitly defines a general notion of discrepancy between distributions:

$$d_S(\mathbb{P}_0, \mathbb{P}_1) = \sup_{\psi_S \in \Psi, \eta \in \Xi} \rho(\psi_S(\eta(\mathbb{P}_1))) + \rho(c - \psi_S(\eta(\mathbb{P}_0))),$$

for some function classes $\Psi$ and $\Xi$, a monotone function $\rho : \mathbb{R} \to \mathbb{R}$, and constant $c$. This generalized discrepancy includes many existing adversarial objectives as special cases. For example, if $\rho$ and $\psi_S$ are identity functions, $\Xi$ the space of 1-Lipschitz functions and $\eta$ an embedding into $\mathbb{R}$, we obtain the 1-Wasserstein distance used by WGAN (Arjovsky et al., 2017; Gulrajani et al., 2017).

Similarly, our two-sample discriminator implicitly defines the following discrepancy:

$$d_{2S}(\mathbb{P}_0, \mathbb{P}_1) = \sup_{\psi_{2S} \in \Psi, \eta \in \Xi} \rho(\psi_{2S}(|\eta(\mathbb{P}_0) - \eta(\mathbb{P}_1)|)), \qquad (4.1)$$

where $\psi, \Xi$ are as before. This form can be thought of as generalizing other discrepancy measures like MMD (Gretton et al., 2005; Fukumizu et al., 2008; Smola et al., 2007), defined as:

$$\text{MMD}^2(U, V) = \left\| \frac{1}{n} \sum_{i=1}^{n} \phi(u_i) - \frac{1}{m} \sum_{j=1}^{m} \phi(v_j) \right\|_2^2,$$

where $\phi(\cdot)$ is some feature mapping, and $k(u, v) = \phi(u)^\top \phi(v)$ is the corresponding kernel function. Letting $\phi$ be the identity function corresponds to computing the distance between the sample means. More complex kernels result in distances that use higher-order moments of the two samples. In adversarial models, MMD and its variants (e.g., central moment discrepancy) have been used either as a direct objective for the target model or as a form of adversary (Zellinger et al., 2017; Sutherland

et al., 2017; Li et al., 2015; Dziugaite et al., 2015). However, most of them require hand-picking the kernel. An adaptive feature function, in contrast, may be able to adapt to the given distributions and thereby discriminate better. Our distributional adversary addresses this drawback and, at the same time, generalizes MMD, owing to the fact that neural networks are universal approximators. Since it is trainable, the underlying witness function evolves as training proceeds, taking a simpler form when the two distributions (generated and true data) are easily distinguishable, and becoming more complex as they start to resemble. In this sense our distributional adversary bears similarity to the recent work by Li et al. (2017). However, their learned sample embedding is used as input to a Gaussian kernel to compute the MMD, inheriting the quadratic time complexity (in sample size) of kernel MMD. Our model uses a neural network on top of the mean embedding differences to compute the discrepancy, resulting in *linear* time complexity and additional flexibility.

A more general family of discrepancy measures of interest in the GAN literature are Integral Probability Metrics (IPM), defined as

$$d_{\mathcal{F}}(\mathbb{P}_0, \mathbb{P}_1) = \sup_{f \in \mathcal{F}} |\mathbb{E}_{x \sim \mathbb{P}_0} f(x) - \mathbb{E}_{x \sim \mathbb{P}_1} f(x)|,$$

where $\mathcal{F}$ is a set of measurable and bounded real valued functions. Comparing this to Eq. (4.1) shows that our two-sample discriminator discrepancy generalizes IPMs too. In this sense, our model is also connected to GAN frameworks based on IPMs (Mroueh & Sercu, 2017; Mroueh et al., 2017).

**Minibatch Discrimination.** Initiated by (Salimans et al., 2016) to stabilize training, a line of work in the GAN literature considers statistics of minibatches to train generative models, known as *minibatch discrimination*. Batch normalization (Ioffe & Szegedy, 2015) can be viewed as a form of minibatch discrimination, and is known to aid GAN training (Radford et al., 2015). Zhao et al. (2017) proposed a repelling regularizer that operates on a minibatch and orthogonalizes the pairwise sample representation, keeping the model from concentrating on only a few modes. We derive our adversaries as acting on full distributions instead of batches, but approximate population expectations with samples in practice. In this sense, our implementation can be viewed as generalizing minibatch discrimination: we leverage neural networks for the design of different forms of minibatch discrimination. Such discrimination does not need hand-crafted objectives as in existing work, and will be able to adapt to the data distribution and target models.

**Permutation Invariant Networks.** The NME (3.2) is a *permutation invariant* operator on (un-ordered) samples. Recent work has explored neural architectures that operate on sets and are likewise invariant to permutations on their inputs. Vinyals et al. (2016) propose a content attention mechanism for unordered inputs of variable length. Later work (Ravanbakhsh et al., 2017; Zaheer et al., 2017) embeds all samples into a fixed-dimensional latent space, and then sums them to obtain a fixed-dimensional vector representation, used as input to another network. Liu et al. (2017) use a similar network for embedding a set of images into a latent space, but aggregate using a (learned) weighted sum. Although the structure of our NEM resembles these networks in terms of permutation invariance, it differs in its motivation—discrepancy measures—as well as its usage within discriminators in adversarial training settings.

**Other Related Work.** Various other approaches have been proposed to address training instability and mode collapse in GANs. Many such methods resort to more complex network architectures or better-behaving objectives (Radford et al., 2015; Huang et al., 2017; Zhang et al., 2016; Che et al., 2017; Zhao et al., 2017; Arjovsky et al., 2017), while others add more discriminators or generators (Durugkar et al., 2017; Tolstikhin et al., 2017) in the hope that training signals from multiple sources lead to more stable training and better mode coverage.

## 5 EMPIRICAL RESULTS

We demonstrate the effectiveness of DAN training by applying it to generative models and domain adaptation.[3] In generative models, we observe remarkably better mode recovery than non-distributional models on both synthetic and real datasets, through both qualitative and quantitative evaluation of generated samples. In domain adaptation, we leverage distributional adversaries to align latent spaces of source and target domains, and see strong improvements over state-of-the-art.

---

[3]Please refer to the appendix for all details on datasets, network architecture, training procedures and results.

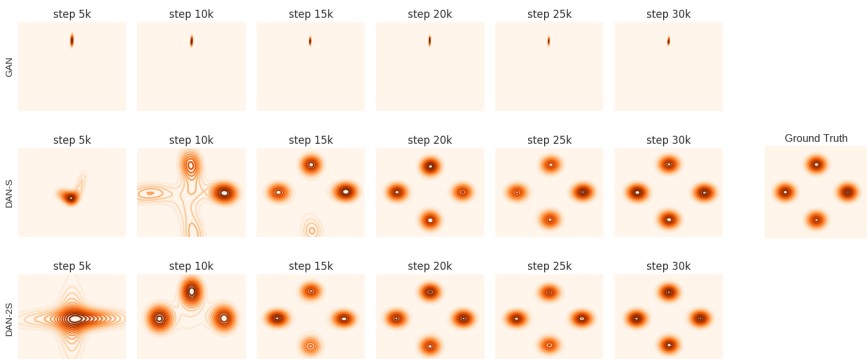

Figure 2: Mixture of 4 Gaussians. GAN training leads to *unrecoverable* mode collapse, with only one of the true distribution's modes being recovered. The two distributional training approaches (bottom 2 rows, $\lambda = 0$) capture all modes, and are able to recover from a missing mode (second column).

## 5.1 SYNTHETIC DATA: MODE RECOVERY IN MULTIMODAL DISTRIBUTIONS

We first test DAN in a simple generative setting, where the true data distribution is a simple two-dimensional mixture of Gaussians. A similar task was used as a proof of concept for mode recovery by Metz et al. (2017). We generate a mixture of four Gaussians with means equally spaced on a circle of radius 6, and variances of 0.02. We compare our distributional adversaries against various discriminator frameworks, including GAN, WGAN (Arjovsky et al., 2017) and WGAN-GP (Gulrajani et al., 2017). We use *equivalent*[4] simple feed-forward networks with ReLU activations for all discriminators. For these synthetic distributions, we use the pure distributional objective for DAN (i.e., setting $\lambda = 0$ in (3.5)). Figure 2 displays the results for GAN and our methods; in the appendix, we show results for other models, different $\lambda$ settings and other synthetic distributions. Overall, while other methods suffer from mode collapse throughout (GAN) or are sensitive to network architectures and hyperparameters (WGAN, WGAN-GP), our DANs consistently recover all modes of the true distribution and are stable across a reasonable range of $\lambda$'s.

## 5.2 MNIST GENERATION: RECOVERING MODE FREQUENCIES

Mode recovery entails not only *capturing* a mode, i.e., generating samples that lie in the corresponding mode, but also recovering the true *probability mass* of the mode. Next, we evaluate our model on this criterion: we train DAN on MNIST and Fashion-MNIST (Xiao et al., 2017), both of which have 10-class balanced distributions. Since the generated samples are unlabeled, we train an external classifier on (Fashion-)MNIST to label the generated data.

Besides the original GAN, we compare to recently proposed generative models: RegGAN (Che et al., 2017), EBGAN (Zhao et al., 2017), WGAN (Arjovsky et al., 2017) and its variant WGAN-GP (Gulrajani et al., 2017), and GMMN (Li et al., 2015). To keep the approaches comparable, we use a similar neural network architecture in all cases, and did not tailor it to any particular model. We trained models without Batch Normalization (BN), except for RegGAN and EBGAN, which we observed to consistently benefit from BN. For DAN, we use the the regularized objective (3.5) with $\lambda > 0$. Both adversaries in this formulation use the same architecture except for the averaging layer of the distributional adversary, and share weights of the pre-averaging layers.

Figure 3 shows the results. Training with the vanilla GAN, RegGAN or GMMN leads to generators that place too much mass on some modes and ignore others, leading to a large TV distance between the generated label distribution and the correct one. EBGAN and WGAN perform slightly worse than WGAN-GP and DAN on MNIST, and significantly worse on the (harder) Fashion-MNIST dataset. WGAN-GP performs on par with DAN on MNIST, and slightly worse than DAN on Fashion-MNIST. Moreover, WGAN-GP is in general more sensitive to hyperparameter selection (Appendix B).

---

[4] The DAN discriminator architecture is identical to the others', differing only in the aggregation layer.

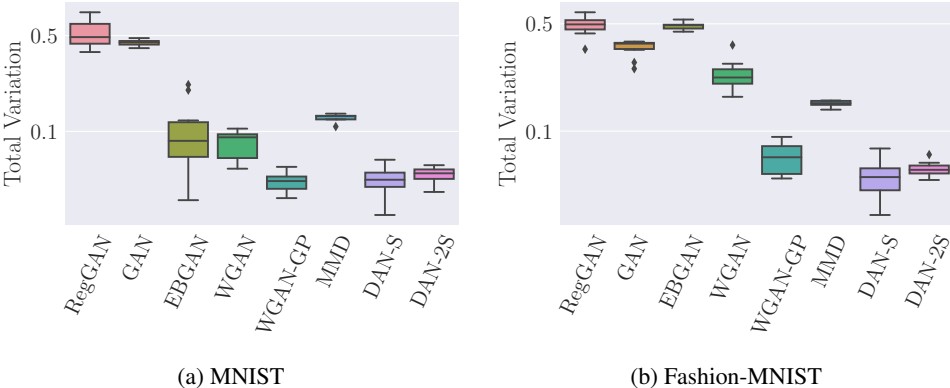

(a) MNIST           (b) Fashion-MNIST

Figure 3: Total variation distances (in log-scale) between generated and true (uniform) label distributions over 5 repetitions. DAN achieves the best and most stable mode frequency recovery.

## 5.3   IMAGE GENERATION: SAMPLE DIVERSITY AS EVIDENCE OF MODE COVERAGE

We also test DAN on a harder task, generating faces, and compare against the samples generated by DCGAN. The generated examples in Figure 4 show that DCGAN+BN exhibits an obvious

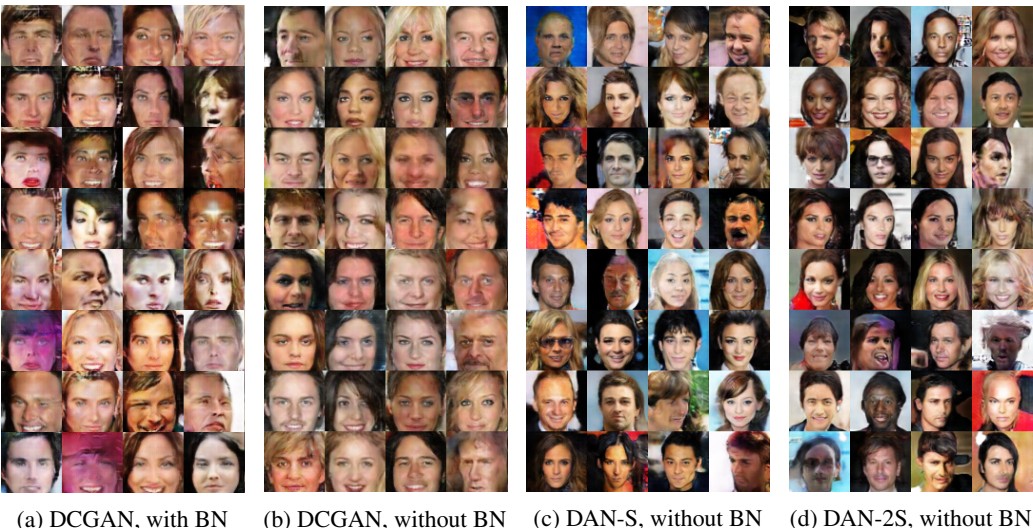

(a) DCGAN, with BN    (b) DCGAN, without BN    (c) DAN-S, without BN    (d) DAN-2S, without BN

Figure 4: Samples generated by DCGAN, DAN-S and DAN-2S trained on the CelebA dataset.

mode-collapse, with most faces falling into a few clusters. DCGAN without BN generates better images, but they still lack feature diversity (e.g., backgrounds, hairstyle). The images generated by DAN, in contrast, exhibit a much richer variety of features, indicating a better coverage of the data distribution's modes.

## 5.4   DOMAIN ADAPTATION

In the last set of experiments, we test DAN in the context of domain adaptation. We compare against DANN (Ganin et al., 2016), an adversarial training algorithm for domain adaptation that uses a domain-classifier adversary to enforce similar source and target representations, thus allowing for the source classifier to be used in combination with the target encoder. We use the DAN framework to further encourage distributional similarities between representations from different domains.

We first compare the algorithms on the *Amazon reviews* dataset preprocessed by Chen et al. (2012). It consists of four domains: books, dvd, electronics and kitchen appliances, each of which contains

Table 1: Results of domain adaptation on Amazon dataset. We report mean prediction accuracy (plus standard deviation) over 5 different runs of each method.

| Source | Target | DANN | DAN-S | DAN-2S |
|---|---|---|---|---|
| books | dvd | 77.1 (1.4) | 77.8 (0.5) | 78.2 (0.8) |
| books | electronics | 74.4 (1.1) | 75.7 (0.4) | 74.8 (1.1) |
| books | kitchen | 77.2 (1.1) | 79.0 (0.4) | 76.9 (0.6) |
| dvd | books | 75.0 (0.8) | 75.4 (0.9) | 74.4 (2.2) |
| dvd | electronics | 75.6 (0.9) | 76.0 (0.9) | 76.3 (1.4) |
| dvd | kitchen | 79.4 (1.0) | 80.4 (0.9) | 78.0 (0.9) |
| electronics | books | 70.3 (1.0) | 72.4 (0.8) | 72.0 (1.9) |
| electronics | dvd | 69.4 (2.6) | 72.4 (2.0) | 72.3 (1.7) |
| electronics | kitchen | 83.6 (0.5) | 85.0 (0.4) | 84.7 (0.7) |
| kitchen | books | 69.5 (0.9) | 70.4 (0.9) | 70.5 (1.4) |
| kitchen | dvd | 69.4 (2.2) | 73.8 (0.8) | 71.5 (2.9) |
| kitchen | electronics | 82.7 (0.5) | 82.5 (0.2) | 83.4 (0.3) |
| avg. imp (over DANN) | | — | **1.41** | 0.92 |

reviews encoded in 5,000-dimensional feature vectors and binary labels indicating positive/negative reviews. Results are shown in Table 1. Our models outperform the GAN-based DANN on most source-target pairs, with an average improvement in accuracy of $1.41\%$ for DAN-S and $0.92\%$ for DAN-2S. Note that we directly integrated DAN without any network structure tuning – we simply set the network structure of the distributional adversary to be the same as that of original discriminator in DANN (except that it takes an average of latent representations in the middle).

Lastly, we test the effectiveness of DAN on a domain adaptation task for image label prediction. The task is to adapt from MNIST to MNIST-M (Ganin et al., 2016), which is obtained by blending digits over patches randomly extracted from color photos from BSDS500 (Arbelaez et al., 2011). The results are shown in Table 2. DAN-S improves over DANN by $\sim 5\%$. Again, these results were obtained by simply plugging in the DAN objective, demonstrating the ease of using DAN.

Table 2: Results of domain adaptation from MNIST to MNIST-M: prediction accuracy averaged over 5 different runs.

| Source Only | Target Only | DANN | DAN-S | DAN-2S |
|---|---|---|---|---|
| 52.2 (2.0) | 95.6 (0.2) | 75.5 (0.9) | **80.6** (2.8) | 79.2 (1.2) |

## 6 DISCUSSION AND FUTURE WORK

In this work, we propose a distributional adversarial framework that, as opposed to common approaches, does not rely on a sum of observation-wise functions, but considers a sample (collection of observations) as a whole. We show that when used within generative adversarial networks, this different approach to distribution discrimination has a stabilizing effect and remedies the well-known problem of mode collapse. One likely reason for this is its ability to share information across observations when computing gradients. The experimental results obtained with this new approach offer a promising glimpse of the advantages of genuine sample-based discriminators over common alternatives that are separable over observations, while the simplicity and ease of implementation make this approach an appealing plug-in, easily compatible with most existing models.

The framework proposed here is fairly general and opens the door for various possible extensions. The two types of discriminators proposed here are by no means the only options. There are many other approaches in the distributional discrepancy literature to draw inspiration from. One aspect that warrants additional investigation is the effect of sample size on training stability and mode coverage. It is sensible to expect that in order to maintain global discrimination power in settings with highly

multimodal distributions, the size of samples fed to the discriminators should grow, at least with the number of modes. Formalizing this relationship is an interesting avenue for future work.

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

# A    TRAINING FOR DAN

We show the complete training procedure for DAN in Algorithm 1 and a visual illustration in Figure 5. Note that we set a step number $k$ such that every other $k$ iterations we update the distributional adversary, otherwise we update the single-observation adversary.

---

**Algorithm 1** Training Procedure for DAN-S/2S.

---

**Input:** total number of iterations $T$, size of minibatch $B$, step number $k$, model mode $\xi \in \{S, 2S\}$
**for** $i = 1$ to $T$ **do**
    Draw samples $X = \{x^{(1)}, \dots, x^{(B)}\} \sim \mathbb{P}_x$, $Z = \{z^{(1)}, \dots, z^{(B)}\} \sim \mathbb{P}_z$
  **if** $mod(i, k) = 0$ and $\lambda_1 > 0$ **then**
    **if** $\xi = S$ **then**
      Update distributional adversary $D_S = \{\psi_S, \eta\}$ with one gradient step on loss:

$$\log(\psi_S(\eta(X))) + \log(1 - \psi_S(\eta(G(Z))))$$

    **else if** $\xi = 2S$ **then**
      Divide $X$ and $Z$ evenly into $X_1$, $X_2$ and $Z_1$, $Z_2$ respectively
      Update distributional adversary $D_{2S} = \{\psi_{2S}, \eta\}$ with one gradient step on loss:

$$\log(\psi_{2S}(|\eta(X_1) - \eta(X_2)|)) + \log(\psi_{2S}(|\eta(G(Z_1)) - \eta(G(Z_2))|)) +$$
$$\log(1 - \psi_{2S}(|\eta(X_1) - \eta(G(Z_2))|)) + \log(1 - \psi_{2S}(|\eta(G(Z_1)) - \eta(X_2)|))$$

    **end if**
  **else if** $\lambda_2 > 0$ **then**
    Update discriminator $D$ with one gradient step on loss:

$$\frac{1}{B}\sum_{i=1}^{B}\left[\log(D(x^{(i)})) + \log(1 - D(G(z^{(i)})))\right]$$

  **end if**
  Draw samples $X = \{x^{(1)}, \dots, x^{(B)}\} \sim \mathbb{P}_x$, $Z = \{z^{(1)}, \dots, z^{(B)}\} \sim \mathbb{P}_z$
  **if** $\xi = S$ **then**
    Update $G$ with gradient step on loss:

$$\frac{1}{B}(\lambda_1 \sum_{i=1}^{B}\log(1 - D(G(z^{(i)})))) + \lambda_2 \log(1 - \psi_S(\eta(G(Z)))))$$

  **else if** $\xi = 2S$ **then**
    Divide $X$ and $Z$ into $X_1$, $X_2$ and $Z_1$, $Z_2$
    Update $G$ with gradient step on loss:

$$\frac{1}{B}(\lambda_1 \sum_{i=1}^{B}\log(1 - D(G(z^{(i)})))) +$$
$$\lambda_2/2 \log(1 - \psi_{2S}(|\eta(X_1) - \eta(G(Z_2))|)) +$$
$$\lambda_2/2 \log(1 - \psi_{2S}(|\eta(G(Z_1)) - \eta(X_2)|))$$

  **end if**
**end for**

---

# B    FURTHER RESULTS ON SYNTHETIC DATASET

We first show the full result for mode recovery when the true data distribution is a mixture of $4$ Gaussians with variances of $0.02$ on a circle of radius $6$. The generator consists of a fully connected network with 3 hidden layers of size 128 with `ReLU` activations, followed by a linear projection to 2 dimensions. The discriminator consists of a fully connected network with 3 hidden layers of size 32 with `ReLU` activations, followed by a linear projection to 1 dimension. Latent vectors are sampled

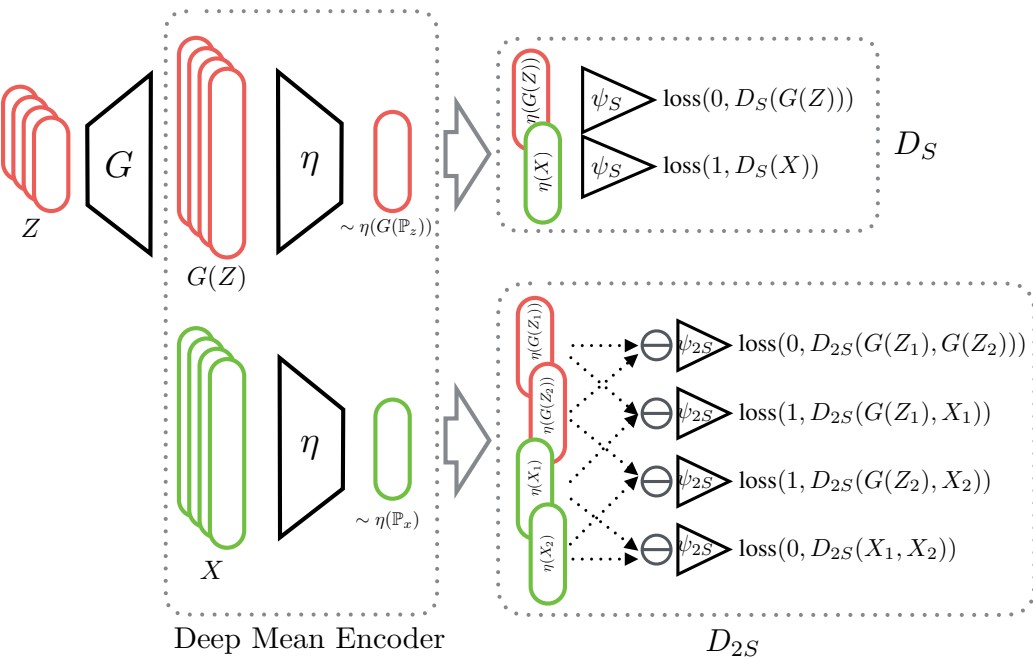

Figure 5: DAN-S and DAN-2S models and corresponding losses, where $X = \{x^{(i)}\}_{i=1}^B \sim \mathbb{P}_x$, $Z = \{z^{(i)}\}_{i=1}^B \sim \mathbb{P}_z$, $X_1 := \{x^{(i)}\}_{i=1}^{\frac{B}{2}}$, $X_2 = \{x^{(i)}\}_{i=\frac{B}{2}+1}^B$, $Z_1 := \{z^{(i)}\}_{i=1}^{\frac{B}{2}}$ and $Z_2 = \{z^{(i)}\}_{i=\frac{B}{2}+1}^B$

uniformly from $[-1, 1]^{256}$. For WGAN, the weight clipping parameter is set to $0.01$. For WGAN-GP, $\lambda$ the weight for the gradient penalty is set to $0.1$. For DAN, the distributional adversaries have two initial hidden layers of size 32, after which the latent representations are averaged across the batch. The mean representation is then fed to a fully connected layer of size 32 and a linear projection to 1 dimension. WGAN is optimized using RMSProp (Tieleman & Hinton, 2012) with learning rate of $5 \times 10^{-5}$. All other models are optimized using Adam (Kingma & Ba, 2014) with learning rate of $10^{-4}$ and $\beta_1 = 0.5$. Minibatch size is fixed to $512$.

The intuitive argument presented in Sec. 2 suggests that GAN training would result in *the same single* mode being recovered throughout, which is also confirmed in the first row of Figure 6: the generator's distribution is concentrated around a single mode, from which it cannot due to its inherited disadvantage of single observation-based discriminator and logistic loss. WGAN does not stuck at a single mode but the mode collapse still happens. On the other hand, WGAN-GP and DAN is able to constantly recover all 4 far-apart modes.

Note that the gradient scaling argument in Sec. 2 only says that points *far* away (i.e. with low probability under $G$) from $G$'s modes will be down weighted. But points close to the boundary of $D$'s decision will have a large $\nabla D(x_G^{(i)})$ term, and thus up-weighting gradients and can cause the $G$'s whole mode to shift towards the boundary. If after this shift, enough of $G$'s mass falls closer to another mode, then this can become the attracting point and further concentrate $G$'s mass. This suggests that, for distributions with spiked, but close modes, $G$'s mass should concentrate but possibly traverse from mode to mode. This effect is indeed commonly observed for the popular 8-gaussians example shown in Figure 7. All other models follow the same argument as in Figure 6.

Though WGAN-GP is able to recover all modes as DAN, we show that the training for WGAN-GP is unstable – a slight change in hyperparameter may end up with totally different generator distribution. We consider the same network architecture as in public available WGAN-GP code[5]: The generator consists of a fully connected network with 3 hidden layers of size 512 with `ReLU` activations, followed by a linear projection to 2 dimensions. The discriminator consists of a fully connected

---

[5]https://github.com/igul222/improved_wgan_training

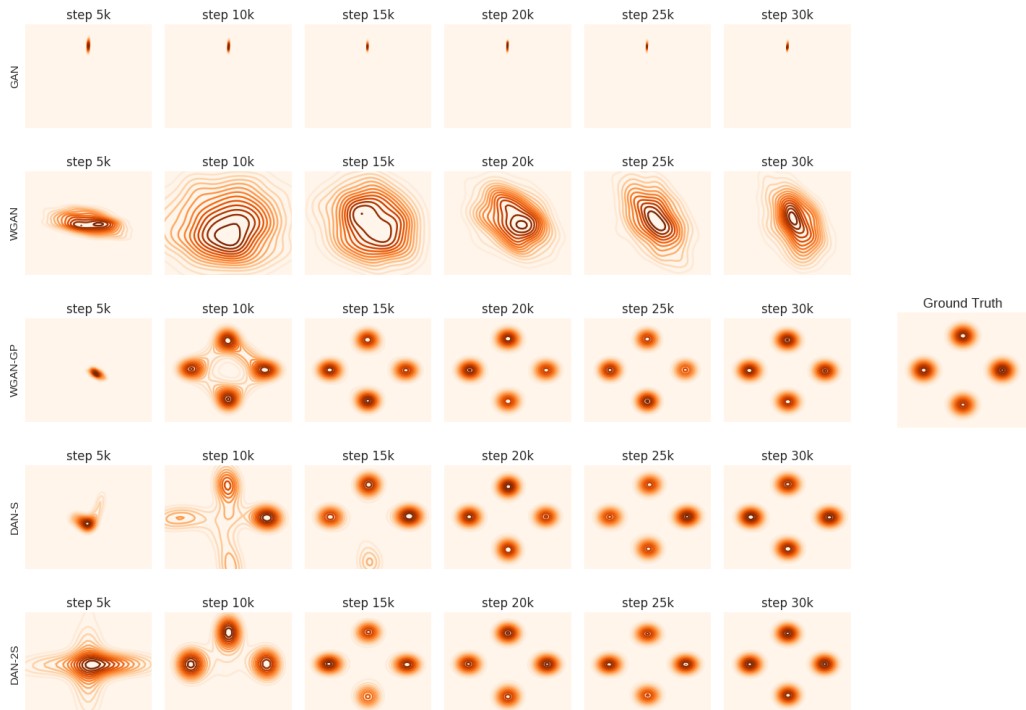

Figure 6: Results for mode recovery on distribution of a mixture of 4 Gaussians on a circle of radius 6. The rightmost plot shows the true data distribution. With GAN training (the 1st row) the generator is only able to capture the same single one of all modes. WGAN (rows 2) do not get stuck at the same single mode since they do not use the logistic loss, but the mode collapse still happens. WGAN-GP and DAN is able to constantly recover all 4 modes.

network with 3 hidden layers of size 512 with `ReLU` activations, followed by a linear projection to 1 dimension. Latent vectors are sampled uniformly from $[-1, 1]^2$. Distributional adversaries starts with 2 hidden layers of size 512, then the latent representations are averaged across the batch. The mean representation is then fed to 1 hidden layer of size 512 and a linear projection to 1 dimension. Minibatch size is fixed to 256.

For the first set of experiments we optimize using Adam (Kingma & Ba, 2014) with learning rate of $10^{-4}$, $\beta_1 = 0.5$ and $\beta_2 = 0.9$, and train the network for $30,000$ iterations, where each iteration consists of 5 updates for discriminator and 1 for generator for WGAN-GP. The result is shown in Figure 8. All models successfully capture all modes, despite their separation. However, a slight change in the optimizer's parameters to $\beta_2 = 0.999$ (the default value) causes WGAN-GP to fail sometime even if we run for $50,000$ iterations, while DAN still constantly recovers the true distribution (cf. Figure 9). We also experiment with WGAN optimized using RMSProp with learning rate of $5 \times 10^{-5}$ under this setting and in contrast to previous experiments where it fails to recover any mode, it is able to recover all modes using this network architecture. This shows that both WGAN and WGAN-GP are sensitive to network structure or hyperparameter settings, while DAN is more stable.

Finally, we show that DAN is stable across a reasonable range of $\lambda$, the trade-off parameter for single observation and distributional adversaries. We set $\lambda = \{0, 0.2, 0.5, 1, 2, 5, 10\}$ for DAN and the results are shown in Figure 10 and 11. While both DAN-S and DAN-2S are stable, DAN-2S is stable in larger range of $\lambda$ than DAN-S.

To summarize, the trends we observe in these these synthetic experiments are: (i) GAN irremediably suffers from mode collapse for this simple dataset, (ii) WGAN(+GP) can fail in recovering modes or are not stable against hyperparameter choices and (iii) DAN constantly recovers all modes and is stable for all reasonable hyperparameter settings.

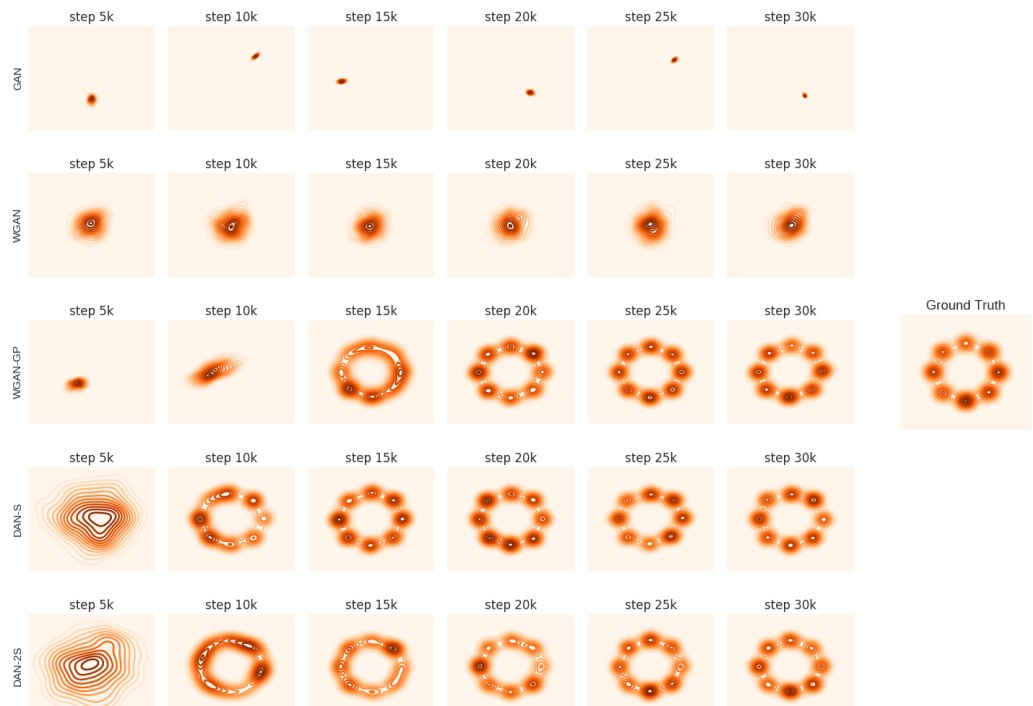

Figure 7: Results for mode recovery on distribution of a mixture of 8 Gaussians on a circle of radius 2. The right-most plot shows the true data distribution. Since modes are closer, the generator in GAN may not get stuck at generating the same single mode, but oscillate between modes, as confirmed by the 1st row. WGAN (rows 2) do not get stuck at the same single mode since they do not use the logistic loss, but the mode collapse still happens. WGAN-GP and DAN is able to constantly recover all 8 modes.

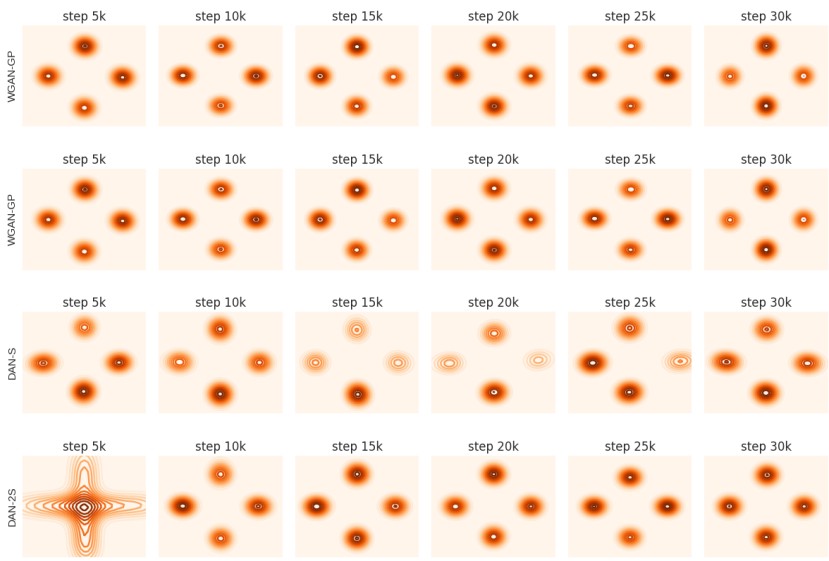

Figure 8: WGAN (row 1) optimized using RMSProp with learning rate of $5 \times 10^{-5}$, WGAN-GP (row 2) and DAN (row 3-4) optimized with Adam with learning rate of $10^{-4}$, $\beta_1 = 0.5$ and $\beta_2 = 0.9$

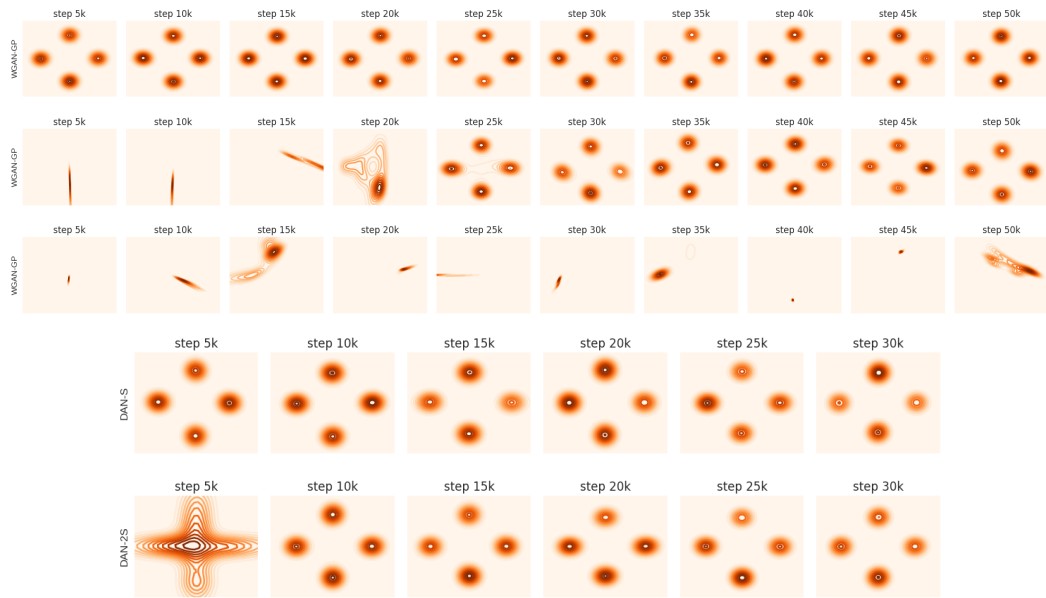

Figure 9: WGAN-GP (rows 1-3) and DAN (rows 4-5) optimized with Adam with learning rate of $10^{-4}$, $\beta_1 = 0.5$ and $\beta_2 = 0.999$. The first 3 rows show random runs for WGAN-GP – it does not constantly recover all modes, even if we run it for longer time.

## C EXPERIMENT DETAILS ON MNIST

We use the same architecture for the generator across all models: three fully-connected layers of sizes $[256, 512, 1024]$ with ReLU activations, followed by a fully connected linear layer and a Sigmoid activation to map it to a 784-dimensional vector. Latent vectors are uniform samples from $[-1, 1]^{256}$. For GAN, RegGAN and DAN the single observation discriminators consist of three fully-connected layers of sizes $[1024, 512, 256]$ with ReLU activations, followed by a fully connected linear layer and Sigmoid activation to map it to a 1-dimensional value. Both adversaries in DAN's use the same architecture except for the averaging layer of the distributional adversary, and share weights of the pre-averaging layers. This is done to limit the number of additional parameters of the regularizer, so as to make training more efficient. The architecture for the decoder in EBGAN is the same with the generator, while the ones for encoders in RegGAN and EBGAN are the same as the discriminator except for the last layer where it maps to 256-dimensional vectors. For GMMN we use a mixture of Gaussian kernels with bandwidths in $\{0.1, 0.5, 1, 5, 10, 50\}$. Throughout the experiments, we set the hyperparameters as shown in Table 3.

| Model | Hypaerparameters |
|---|---|
| RegGAN | $\lambda_1 = \lambda_2 = 0.01 \in \{0.01, 0.1, 1\}$ |
| EBGAN | $m = 0.2 \in \{0.2, 1, 5\}$ |
| WGAN | $n_{\text{critic}} = 2$, Clip value $0.01$ |
| WGAN-GP | $n_{\text{critic}} = 2$, Gradient penalty weight $10$ |
| DAN-S | $k = 5, \lambda = 2 \in \{1, 2, 5\}$ |
| DAN-S | $k = 5, \lambda = 5 \in \{1, 2, 5\}$ |

Table 3: Hyperparameter settings, numbers in curly braces are values tried with grid search.

We use Adam with learning rate of $0.0005$ and $\beta_1 = 0.5$, a fixed minibatch size of 256 and 100 epochs of training. For models except WGAN and DAN we train by alternating between generator and adversary (and potentially encoder and decoder) updates. Generated digits by each model are shown in Figure 12, where we observe a clear advantage of DAN-S and DAN-2S over all other baselines in terms of mode coverage and generation quality.

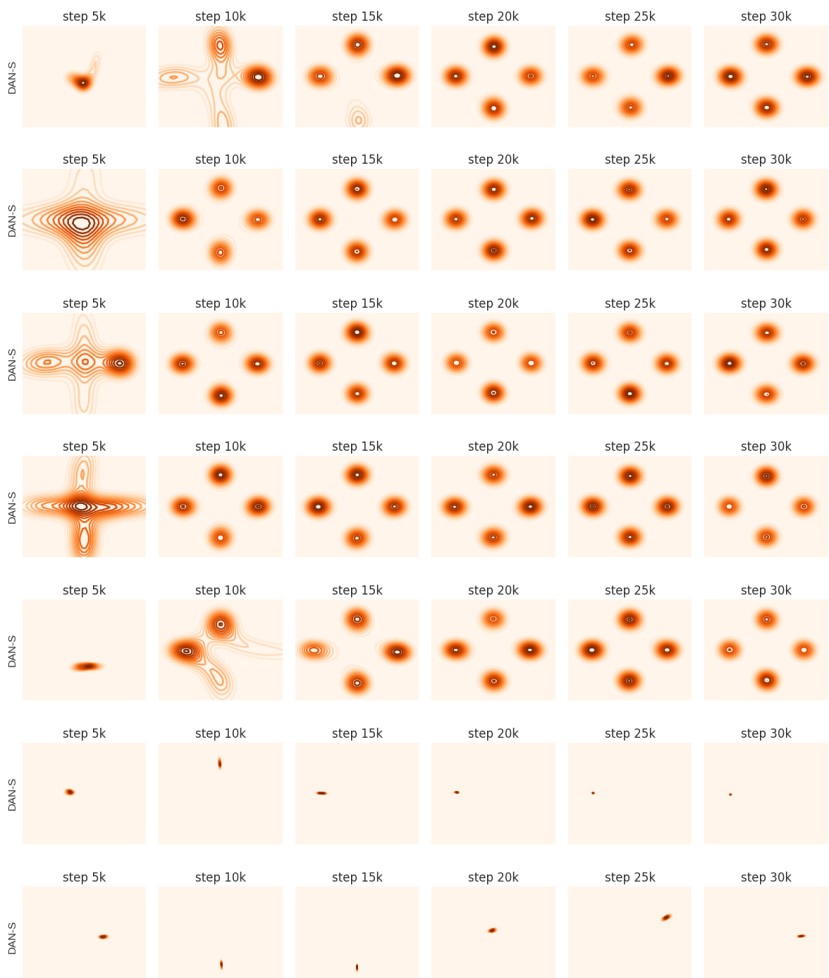

Figure 10: DAN-S with $\lambda$ set to $\{0, 0.2, 0.5, 1, 2, 5\}$ (top-down order).

## D  EXPERIMENT DETAILS ON FASHION-MNIST

We use the same experimental settings as in Section C. Generated examples by each model are shown in Figure 13, where we observe a clear advantage of DAN-S and DAN-2S over all other baselines in terms of mode coverage and generation quality.

### D.1  SENSITIVITY TO BATCH-SIZE

We vary the batch-size in DAN's to see how it affects model performance. Results are shown in Figure 14. We observe that both methods suffer from too small or too large batch-sizes. However, there is a clear distinction between the two: while DAN-S outperforms DAN-2S with smaller batch-sizes, this trend reverses for larger batch-sizes, where DAN-2S achieves better performance and is more stable across repetitions.

## E  EXPERIMENT DETAILS ON SVHN

We present here additional experiments on the SVHN dataset. We compare DAN's against GAN, WGAN, WGAN-GP and GMMN. We use the same architecture for the generator across all models: a latent vector first goes through two fully connected layers with number of hidden units 1,024 and 8,192 respectively. Then it is reshaped to sizes $[8 \times 8 \times 128]$ and goes through 2 transposed

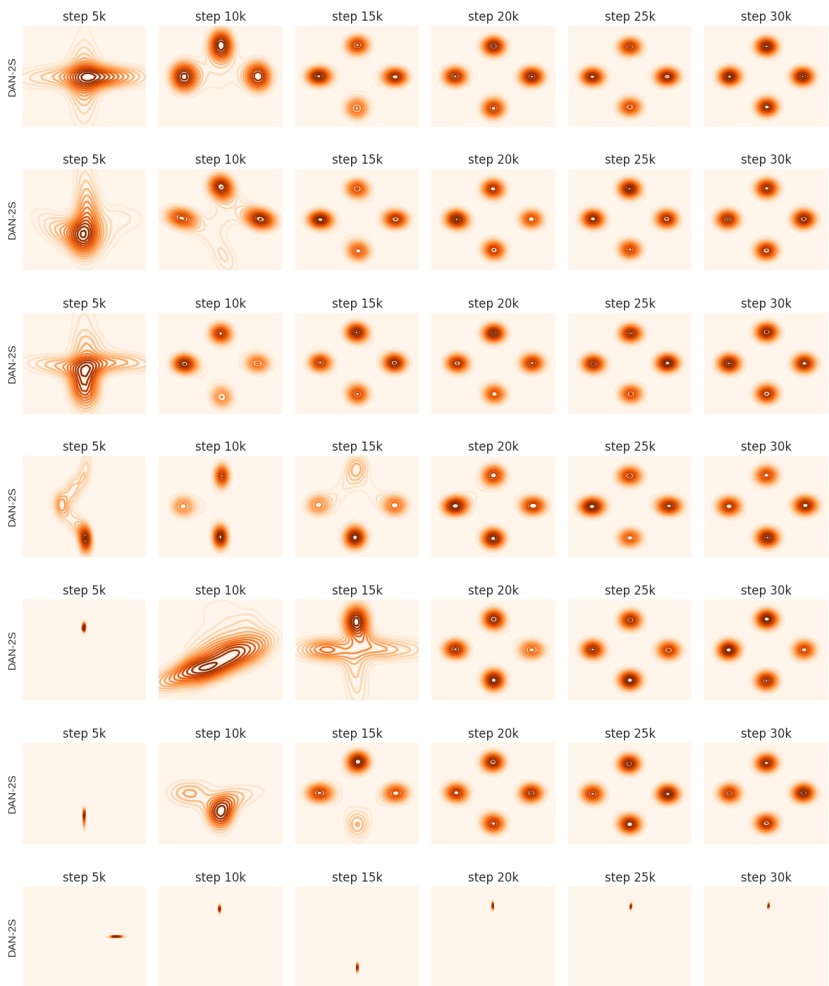

Figure 11: DAN-2S with $\lambda$ set to $\{0, 0.2, 0.5, 1, 2, 5, 10\}$ (top-down order).

convolutional layers with kernel size of 4 and stride 2, followed by a `Sigmoid` activation to map it to a $[32 \times 32 \times 3]$-dimensional vector. Latent vectors are uniform samples from $[-1, 1]^{256}$. For GAN, WGAN variants and DAN the single observation discriminators consist of 2 convolutional layers of kernel size 4 and stride 2, followed by 2 fully connected linear layer with number hidden unites 8,192 and 1,024 respectively, and a `Sigmoid` activation to map it to a 1-dimensional value. Both adversaries in DAN use the same architecture except for the averaging layer of the distributional adversary, and share weights of the pre-averaging layers. For both generators and discriminators we also add batch-normalization layers for all but DAN models. Other hyperparameters and training dynamics are the same as in Section C.

We again compute the generated label distribution and compare it against true (uniform) distribution. TV distances are shown in Figure 15. DAN achieves one of the best and most stable mode frequency recovery. While GMMN performs slightly better than DAN, it suffers from poor generating quality. Specifically, we show generated figures in Figure 16, where DAN's present one of the best mode coverage and generation quality among all.

## F    EXPERIMENT DETAILS ON CIFAR10

We use the same experimental settings as in Section E and run all models on CIFAR10. Generated samples from various models are shown in Figure 17. Again DAN presents one of the best mode coverage and generation quality.

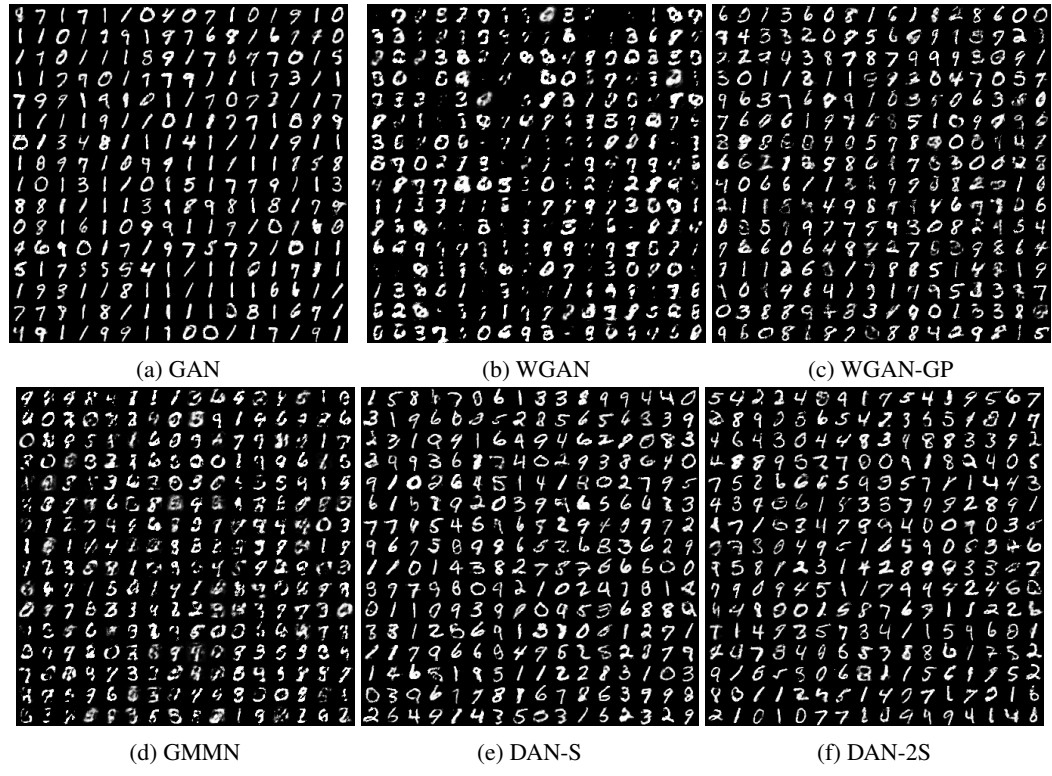

(a) GAN
(b) WGAN
(c) WGAN-GP

(d) GMMN
(e) DAN-S
(f) DAN-2S

Figure 12: GAN, WGAN, WGAN-GP, DAN-S and DAN-2S on MNIST dataset.

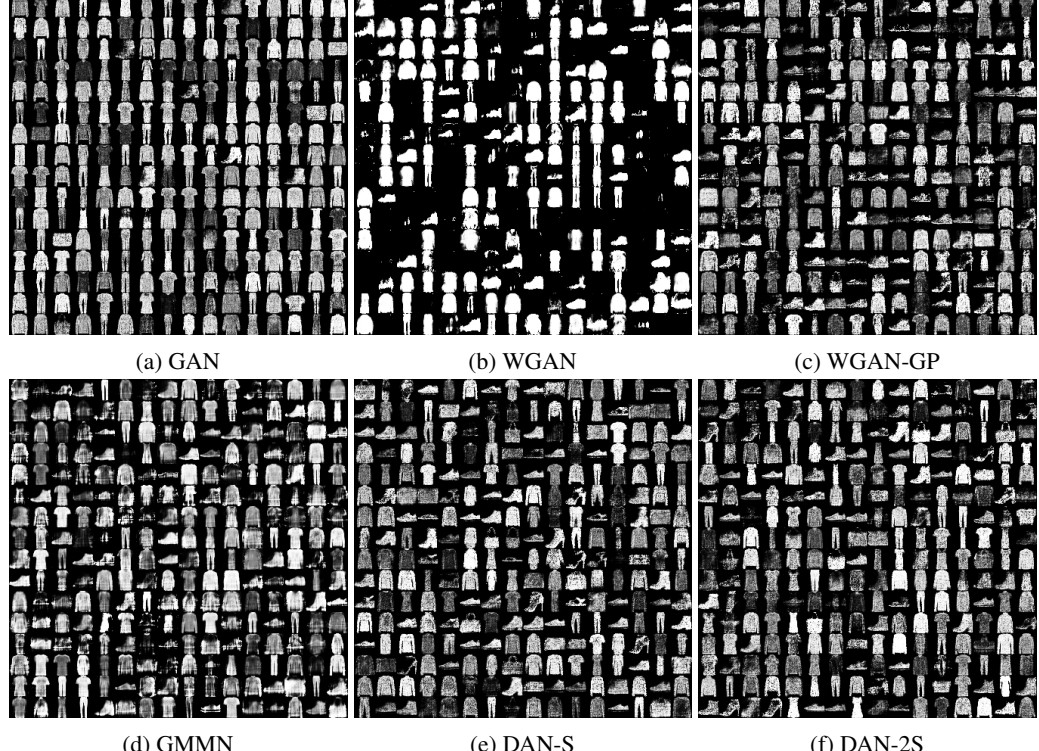

(a) GAN
(b) WGAN
(c) WGAN-GP

(d) GMMN
(e) DAN-S
(f) DAN-2S

Figure 13: GAN, WGAN, WGAN-GP, DAN-S and DAN-2S on Fashion MNIST dataset.

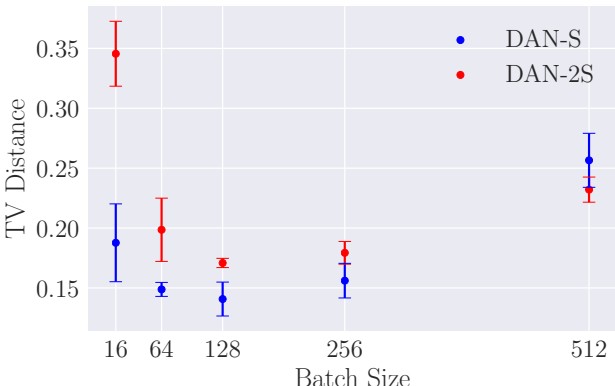

Figure 14: Performances (measured as TV distance between generated and true class distributions) on `fashion-mnist` of DAN's with varying batch-sizes in $\{16, 32, 64, 128, 256, 512\}$. DAN-S is more stable with small batch-sizes while DAN-2S is more stable and achieves better overall performance in the large batch-size regime.

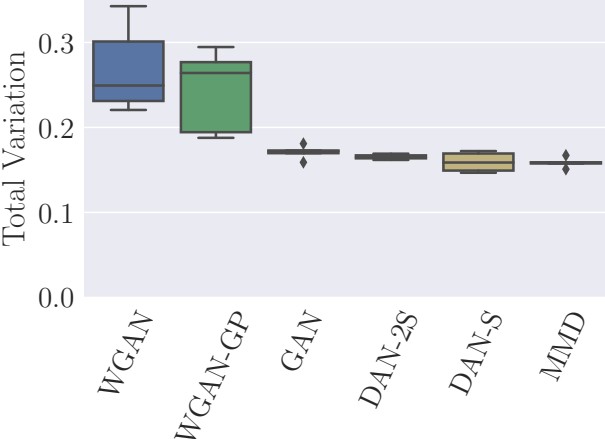

Figure 15: Total variation distances between generated and true (uniform) label distributions over 5 repetitions. Performances are sorted in increasing order, i.e., better performing models stay on the right. DAN achieves one of the best and most stable mode frequency recovery. While GMMN performs slightly better than DAN, it suffers from poor generating quality as seen in Figure 16

## G  EXPERIMENT DETAILS ON CELEBA

We use a publicly available implementation[6] of DCGAN. The network architecture is kept as in the default configuration. We preprocess the image data by first cropping each image to $160 \times 160$ (instead of $108 \times 108$ in default setting) and then resize them to $64 \times 64$. The generator consists of a fully connected linear layer mapping from latent space of $[-1, 1]^{100}$ to dimension 8,192, followed by 4 deconvolution layers, three with `ReLU` activations and the last one followed by `tanh`. The discriminator is the "reverse" of generator, except that the activation function is `Leaky ReLU` the last layer being a linear mapping and a `Sigmoid` activation to 1D value. Both adversaries in DAN use the same architecture except for the averaging layer of the distributional adversary, and share weights of the pre-averaging layers.

---

[6]`https://github.com/carpedm20/DCGAN-tensorflow`

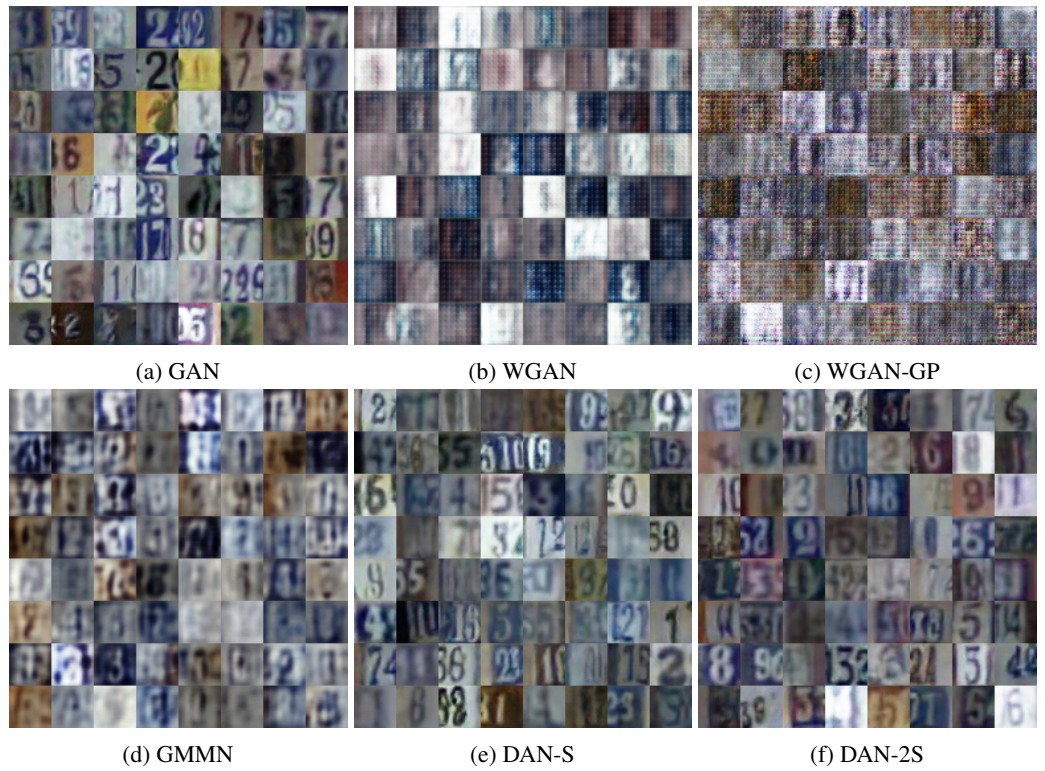

Figure 16: GAN, WGAN, WGAN-GP, GMMN, DAN-S and DAN-2S on the SVHN dataset.

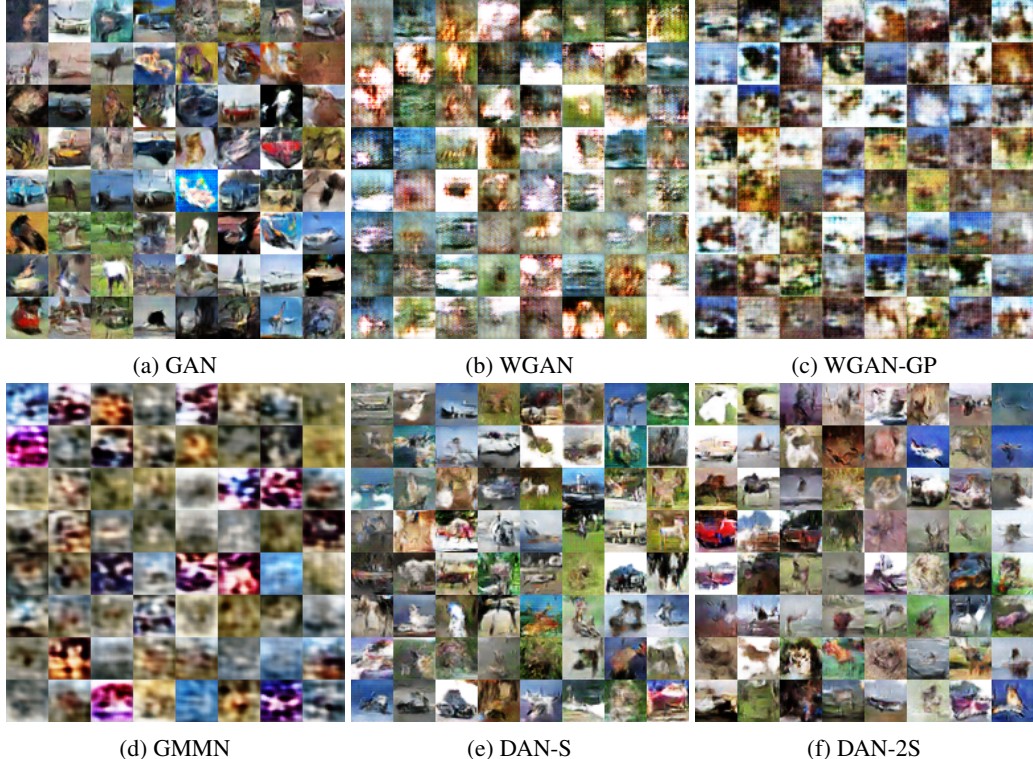

Figure 17: GAN, WGAN, WGAN-GP, GMMN, DAN-S and DAN-2S on the CIFAR10 dataset.

We use Adam with learning rate of $5 \times 10^{-4}$ and $\beta_1 = 0.5$ for training. The batch size is fixed to 64 and we train the network for 30 epochs. For DAN-$\xi$, we set $\lambda = 5$ and $k = 5$. Samples generated by different models are shown in Figure 18.

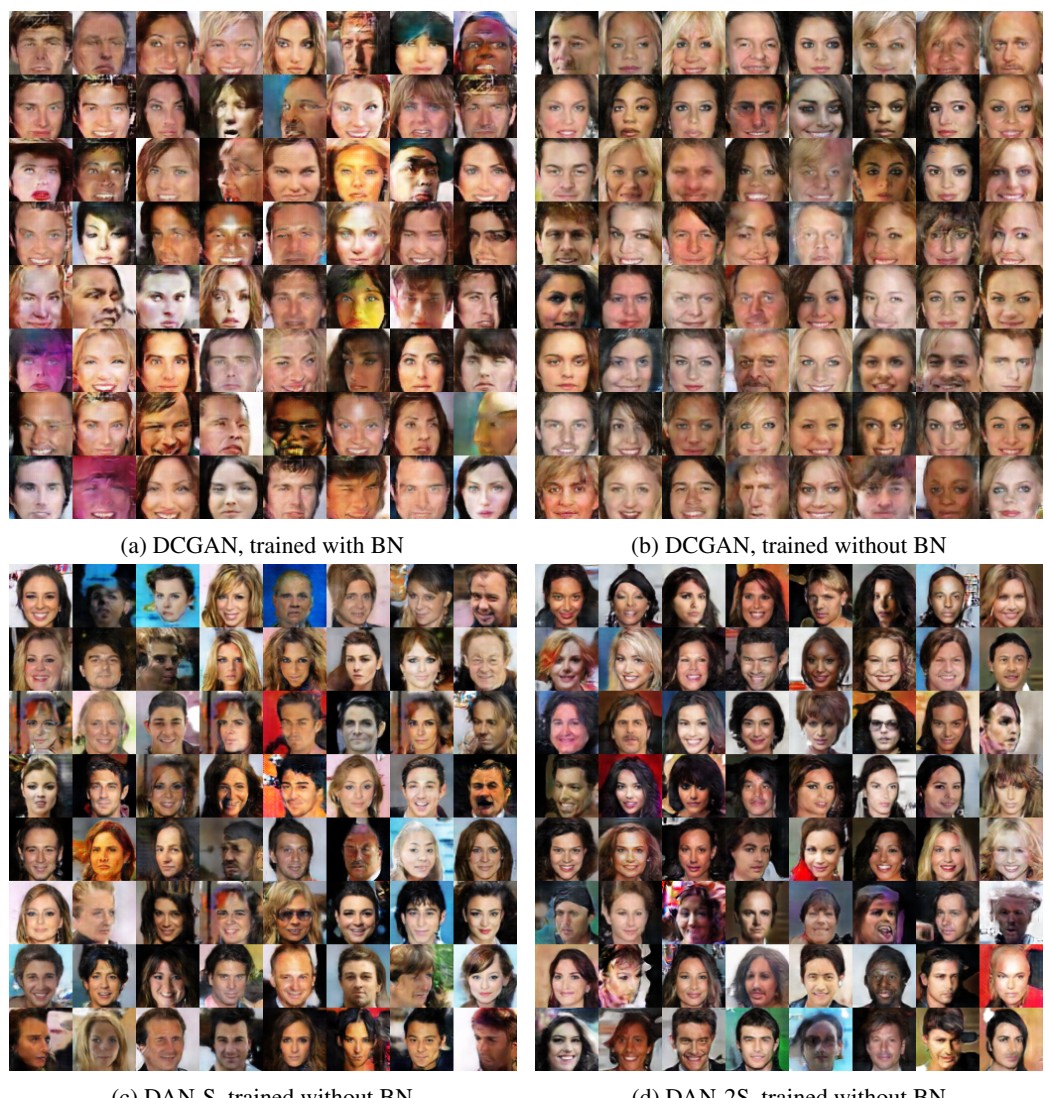

(a) DCGAN, trained with BN  (b) DCGAN, trained without BN

(c) DAN-S, trained without BN  (d) DAN-2S, trained without BN

Figure 18: DCGAN, DAN-S and DAN-2S trained on the CelebA dataset. Both DAN-S and DAN-2S demonstrate much more diversity in generated samples compared to DCGAN.

## H  EXPERIMENT DETAILS FOR DOMAIN ADAPTATION

We adapt the publicly available code[7] for DANN to train on the Amazon and MNIST-M datasets. For the former, we let the encoder of source/target domains consists of 3 fully connected layers of sizes $1000, 500, 100$ with `ReLU` activations, followed by a projection to 2 dimensions for both adversaries and classifier. For the latter dataset we set the model structure to be default in the original code: the encoder of source/target domains consists of 2 convolutional layers, each followed by `ReLU` and MaxPooling layers. The encoded vector is further mapped to 100-dimensional vector followed by `ReLU`, and finally 2-dimensional vector for adversaries and 10-dimensional vectors for classifier. We set the weights for the single observation adversary and distributional adversary to be both 0.1 and both 1 for Amazon dataset and MNIST-M dataset respectively.

---

[7] https://github.com/pumpikano/tf-dann

