# OpenReview forum: "Distributional Adversarial Networks"
_ICLR.cc/2018/Conference — Invite to Workshop Track_

### Official Review · AnonReviewer1 · 2017-11-26
**interesting ideas, would like more empirical support**

**Rating:** 6
**Confidence:** 3

**Review:**

The paper proposes to replace single-sample discriminators in adversarial training with discriminators that explicitly operate on distributions of examples, so as to incentivize the generator to cover the full distribution of the training data and not collapse to isolated modes.

The idea of avoiding mode collapse by providing multiple samples to the discriminator is not new; the paper acknowledges prior work on minibatch discrimination but does not really describe the differences with previous work in any technical detail. Not being highly familiar with this literature, my reading is that the scheme in this paper grounds out into a somewhat different architecture than previous minibatch discriminators, with a nice interpretation in terms of a sample-based approximation to a neural mean embedding. However the paper does not provide any empirical evidence that their approach actually works better than previous approaches to minibatch discrimination. By comparing only to one-sample discriminators it leaves open the (a priori quite plausible) possibility that minibatch discrimination is generally a good idea but that other architectures might work equally well or better, i.e., the experiments do not demonstrate that the MMD machinery that forms the core of the paper has any real purchase.

The paper also proposes a two-sample objective DAN-2S, in which the discriminator is asked to classify two sets of samples as coming from the same or different distributions. This is an interesting approach, although empirically it does not appear to have any advantage over the simpler DAN-S -- do the authors agree with this interpretation? If so it is still a worthwhile negative result, but the paper should make this conclusion explicit. Alternately if there are cases when the two-sample test is actually recommended, that should be made explicit as well.

Overall this paper seems borderline -- a nice theoretical story, grounding out into a simple architecture that does seem to work in practice (the domain adaptation results are promising), but with somewhat sloppy writing and experimentation that doesn't clearly demonstrate the value of the proposed approach. I hope the authors continue to improve the paper by comparing to other minibatch discrimination techniques. It would also be helpful to see value on a real-world task where mode collapse is explicitly seen as a problem (and/or to provide some intuition for why this would be the case in the Amazon reviews dataset).

Specific comments:
- Eqn (2.2) is described as representing the limit of a converged discriminator, but it looks like this is just the general gradient of the objective --- where does D* enter into the picture?
- Fig 1: the label R is never explained; why not just use P_x?
- Section 5.1 "we use the pure distributional objective for DAN (i.e., setting λ != 0 in (3.5))" should this be λ = 0?
- "Results" in the domain adaptation experiments are not clearly explained -- what do the reported numbers represent? (presumably accuracy(stddev) but the figure caption should say this). It is also silly to report accuracy to 2 decimal places when they are clearly not significant at that level.

---

> ### Author Response · Authors · 2018-01-05
> **Response to Reviewer 1**
>
> Comparisons to methods based on minibatch discrimination:
>     For a discussion on conceptual differences, please refer to the general comments. On the experimental side, we have already included comparisons against methods based on minibatch discrimination. RegGAN, one of the methods that we have compared to (Fig. 3), is already operating on a sample by including a “pull-away” loss within minibatches. The comparisons show that DAN’s are superior in performance. In the revised version, we have included a new comparison to GMMN [1], a method based on MMD. The detailed experimental setting is described in Appendix C and E. The results show that our DAN framework is very competitive (on SVHN), and very often better (on MNIST/Fashion-MNIST) than GMMN.
>
> Comparisons between DAN-S and DAN-2S:
>     Our experimental results show that there is no clear-cut winner between them. While in cases of MNIST, Fashion-MNIST and SVHN, DAN-S obtains slightly better mean results, DAN-2S is much more stable and has less variance in performances. There are also some other important differences worth pointing out:
>     1) DAN-2S is more stable over different lambda parameters (Fig. 10 vs Fig. 11).  We observed it to be less prone to mode collapse in simple settings.
>     2) The two methods have distinctly different training dynamics (e.g. Fig. 6 in Appendix A).
>     3) The (new) results on batch-size show that DAN-2S is more robust with larger batch-sizes, while DAN-S is more robust with smaller batch-sizes.
>     4) DAN-S is less computationally expensive.
> Owing to these differences, we decided to keep both methods in the paper, since they have properties which might be appealing in different applications.
>
> Eq 2.2:
>     We need to use the optimal discriminator D* to characterize explicitly the form of the weighting term (namely, the denominator). If D is close to D*, then we know low D implies low P(x). That's the only part where we use it.
>
> All the additional comments and suggestions have been included in the revised version.
>
> [1] Generative moment matching networks. Li, Yujia and Swersky, Kevin and Zemel, Rich. ICML-15

---

### Official Review · AnonReviewer3 · 2017-11-27
**Good and clear paper, well-written, but potentially too incremental**

**Rating:** 6
**Confidence:** 4

**Review:**

I really enjoyed reading this paper. Very well-grounded on theory of two-sample tests and MMD and how these ideas can be beneficially incorporated into the GAN framework. It's very useful purely because of its theoretical support of minibatch discrimination, which always seemed like a bit of a hack. So, excellent work RE quality and clarity. One confusion I had was regarding the details of the how the neural embedding feature embedding/kernel \phi(x) is trained in practice and how it affects performance -- must be extremely significant but certainly under-explored.

I think it's fair to say that the whole paper is approximately minibatch-discrimination + MMD. While this is a very useful and a  much more principled combination, supported by good experimental results, I'm not 100% sure if it is original enough.

I agree with the authors that discriminator "overpowering" of generators is a significant issue and perhaps a little more attention ought to have been given for the generators being more effective w.r.t. 2S tests and distributional discrimination, as opposed to the regularization-based "hack".

I would've also liked to have seen more results, e.g. CIFAR10 / SVHN. One of the best ways to evaluate GAN algorithms is not domain adaptation but semi-supervised learning, and this is completely lacking in this paper.

Overall I would like to see this paper accepted, especially if some of the above issues are improved.

---

> ### Author Response · Authors · 2018-01-05
> **Response to Reviewer 3**
>
> Training of NME:
>     The Neural Mean Embedding (NME) module is trained simultaneously with all the rest of the network. The distinction between the NME and the rest of the adversary is conceptual, made here to emphasize the fact that NME is shared across elements in the sample. In practice, the model is trained end-to-end as one network.
>
> Relation to minibatch discrimination / MMD:
>     Although the implementation of our approach might resemble a combination of  minibatch discrimination and MMD, this analogy does not extend to the motivation, theoretical grounding, adaptability nor the performance of our approach. The methods proposed here do not arise as a post-hoc decision to combine these two aspects, but is rather driven by theoretical insights from two-sample tests and analysis of mode-collapse shown in Sec. 2 & 3. This leads to a general, unifying framework that is grounded on an extensive body of literature on two-sample tests. Further, we go beyond both minibatch discrimination and MMD by learning the minibatch losses and kernels in a data-driven way, while in both minibatch discrimination and MMD one must hand-craft all losses / kernels.
>
> Choice of loss function:
>     Our goal for this work is to motivate theoretically the method and empirically demonstrate its usefulness. While we concede that other approaches to mitigate the “overpowering effect” are very interesting avenues, we employ the regularization-based approach due to its simplicity.
>
> Extra experiments:
>     Please refer to the updated manuscript, Appendix E and F, for results on SVHN and CIFAR10, which show that DAN’s are among the top-performing models in terms of mode coverage and generation quality in these additional tasks too.

---

### Official Review · AnonReviewer2 · 2017-11-27
**Improving GAN training comparing samples instead of observations**

**Rating:** 6
**Confidence:** 3

**Review:**

This paper looks at the problem of mode collapsing when training GANs, and proposes a solution that uses discriminators to compare distributions of samples as opposed to single observations. Using ideas based on the Maximum Mean Discrepancy (MMD) to compare distributions of mini-batches, the authors generalize the idea of mini batch discrimination. Their tweaked architecture achieves nice results on extensive experiments, both simulated and on real data sets.

The key insight is that the training procedure usually used to train GANs does not allow the discriminator to share information across the samples of the mini batch. That is, the authors write out the gradient of the objective and show that the gradient with respect to each observation in the batch is multiplied by a scaling factor based on the discriminator, which is then summed across the batch. As a result, this scaling factor can vanish in certain regions and cause mode collapsing. Instead, the authors look at two different discrepancy metrics, both based on what they call neural mean embeddings, which is based on MMD. After describing them, they show that these discriminators allow the gradient weights to be shared across observations when computing gradients, thus solving the collapsing mode problem. The experiments verify this.

As the authors mentioned, the main idea is a modification of mini batch discrimination, which was also proposed to combat mode collapsing. Thus, the only novel contributions come in Section 3.1, where the authors introduce the mini-batch discriminators. Nevertheless, based on the empirical results and the coherence of the paper (along with the intuitive gradient information sharing explanation), I think it should be accepted.

Some minor points:
-How sensitive is the method to various neural network architectures, initializations, learning rates, etc? I think it's important to discuss this since it's one of the main challenges of training GANs in general.
-Have you tried experiments with respect to the size of the mini batch? E.g. at what mini batch size do we see noticeable improvements over other training procedures?
-Have you tried decreasing lambda as the iteration increases? This might be interesting to try since it was suggested that distributional adversaries can overpower G particularly early in training.
-Figure 1 is interesting but it could use better labelling (words instead of letters)

Overall:
Pros: Well-written, good empirical results, well-motivated and intuitively explained
Cons: Not particularly novel, a modification of an existing idea, more sensitivity results would be nice

---

> ### Author Response · Authors · 2018-01-05
> **Response to Reviewer 2**
>
> Our contributions:
>     Besides the introduction of the adversarial framework in 3.1, other conceptual contributions are: the way to set up the adversarial objective (Sec 3.1) and the analysis of mode collapse in Sec 2 & 3. Differentiation of our work from minibatch discrimination and MMD is mentioned in our general comments.
>
> Sensitivity to learning rates and architectures:
>     Experiments on sensitivity to learning rates have been included in Appendix B. We did not devote more space to other sensitivity results since our main goal was to control for these sources of variability (i.e. fix them across models) and focus on the phenomenon which our model is attempting to solve: mode collapse. Delving into loss sensitivity issues, though interesting, falls outside the scope of this paper.
>
> Sensitivity to batch-size:
>     Following the reviewer’s suggestion about sensitivity results, we included additional experiments on the impact of batch-size on performance (Appendix D). The results on varying batch-sizes show that DAN-2S is more robust with larger batch-sizes, while DAN-S is more robust with smaller batch-sizes.
>
> Fig. 1:
>     We have regenerated it with more explicit labels.

---

### Author Response · Authors · 2018-01-05
**General Comments**

We thank the anonymous reviewers for their thorough feedback. We apologize for the delay in this response; properly addressing their concerns required multiple additional experiments. We believe, however, that the results from these experiments, combined with the responses below, address all of their suggestions and concerns.

First, we address two general points raised by the reviewers:

Differentiation from minibatch discrimination / MMD:
    We propose a new distributional adversarial training framework that is driven by theoretical insights of two-sample tests; this setup makes our framework conceptually very different from minibatch discrimination. Moreover, minibatch discrimination requires hand-crafting of minibatch losses, which may be non-trivial and data-dependent. On the other hand, in the DAN framework, the form of the loss between sets of examples (i.e., samples) is parametrized via a neural network, and is thus adaptive to datasets.
    We concede that MMD and DAN bear similar intuitions, but their implementation is very different. MMD-based methods require hand-crafted kernels with hyperparameters that have to be pre-defined via cross-validation. Our framework goes beyond MMD by parameterizing the kernel with a neural network and learning the kernel in a data-driven way, thus is more expressive and adaptive. Beyond this connection, our framework goes further and generalizes the mean square loss in MMD with another neural network, greatly enriching the adaptivity of the model.
    Empirically, our distributional adversarial framework leads to more stable training and significantly better mode coverage than common single-observation methods. Moreover, it is competitive with --and very often better than-- methods based on minibatch discrimination and MMD (See Fig. 3, 12, 13, 15, 16 and 17).

Other network architecture/loss function/training dynamics:
    Some reviewers suggested changing the network architecture or training dynamics to further boost the performance of our models. While we agree that these directions are promising, our goal for this work is to motivate theoretically the “distributional approach” of our method and to empirically demonstrate its usefulness. We leave further improvement of network architecture and training dynamics as an interesting avenue of future work.

In the revised version, we have:
    1) Added experiments on additional datasets: SVHN, CIFAR10 (Appendices E and F).
    2) Included Generative Moment Matching Networks [1] into the comparison (Section 5.2).
    3) Added experiments showing performance as a function of batch-size for both DAN-S and DAN-2S (Appendix D).
    4) Modified various additional minor issues and typos pointed out by the reviewers.

---

### Decision · Program_Chairs · 2018-01-29
**ICLR 2018 Conference Acceptance Decision**

**Decision:**

Invite to Workshop Track

**Comment:**

All the reviewers and myself have concerns about the potentially incremental nature of this work. While I do understand that the proposed method goes beyond crafting minibatch losses, and instead parametrizes things via a neural network, ultimately it's roughly very similar to simply combining MMD and minibatch discrimination and "learning  the kernel". The theoretical justifications are interesting, but the results are somewhat underwhelming (as an example, DANN's are by no means the state of the art on MNIST->MNIST_M, and this task is rather contrived; the books dataset is not even clearly used by anyone else).

The interesting analysis may make it a good candidate for the workshop track, so I am recommending that.